# Injuries, Pain, and Catastrophizing Level in Gymnasts: A Retrospective Analysis of a Cohort of Spanish Athletes

**DOI:** 10.3390/healthcare10050890

**Published:** 2022-05-12

**Authors:** Andreu Sastre-Munar, Antonia Pades-Jiménez, Natalia García-Coll, Jesús Molina-Mula, Natalia Romero-Franco

**Affiliations:** 1Nursing and Physiotherapy Department, University of the Balearic Islands, E-07122 Palma de Mallorca, Spain; andreusastremunar@hotmail.com (A.S.-M.); antonia.pades@uib.es (A.P.-J.); ngarciacoll@gmail.com (N.G.-C.); natalia.romero@uib.es (N.R.-F.); 2Sport High Performance Centre of Balearic Islands, E-07009 Palma de Mallorca, Spain; 3Health Research Institute of the Balearic Islands (IdISBa), E-07120 Palma de Mallorca, Spain

**Keywords:** gymnastics, musculoskeletal pain, injuries, catastrophization

## Abstract

Physical and psychological demands from gymnastics increase the occurrence of injuries and pain among athletes, whose consequent level of catastrophizing could affect rehabilitation and performance. Although the characteristics of each gymnastics discipline may be key factors, they remain unclear. This study aimed to describe injuries, pain, and catastrophizing levels of gymnasts, according to their discipline and training characteristics. A total of 160 gymnasts fulfilled an online survey at the end of the 2021 season. Eighty gymnasts sustained 106 injuries (mainly ankle), and 128 had current pain (mainly low back). Although results were similar among disciplines, rhythmic gymnasts had a higher prevalence of low back pain (*p* = 0.003) and artistic wrist pain (*p* = 0.011). Gymnasts who sustained an injury displayed higher hours of training (*p* = 0.026), and those with current pain had more sports experience (*p* = 0.001) and age (*p* < 0.001). A higher catastrophizing level was observed in injured gymnasts and correlated with pain level (*p* < 0.001). No other differences were found (*p* > 0.05). Pain and injury prevalence is extremely high among gymnasts, being specific to the gymnastics discipline and increasing catastrophizing experience. Hours of training, age, and sports experience are key related factors, regardless of discipline.

## 1. Introduction

The high psychological demands and physical stress of gymnastics make it the sport with the second highest injury rate [1] and a high prevalence of pain [2]. Although the lower limb is the anatomical location where most injuries occur, they may be different according to gender, level of competition, or training characteristics [1]. Despite the fact that this information is not available in all studies, high injury rates and pain existence are always observed.

The normalization of pain or injuries is a traditional attitude in sports that worsens injuries and decreases sports performance among gymnasts [3,4]. Since this pain experience and these injuries may compromise the sports performance of athletes, they tend to feel catastrophism as an exaggerated negative appreciation of pain perception [5,6].

The consequent situation may affect injury rehabilitation and even the capability of performance [7,8,9], being needed to detect factors related to catastrophism and fix or decrease it. Previous studies have demonstrated associations between central pain factors such as anxiety, pain catastrophizing, and injuries in several sports, including artistic gymnastics [3,6,10]. However, these potential factors seem to depend on the competitive discipline [11].

The factors related to training characteristics or discipline and the relationship with pain catastrophizing, pain, or injuries among gymnasts remain unclear due to a lack of studies [12,13]. A high training volume and a constant state of fatigue are frequent factors that prevent the complete recovery of gymnasts [14,15]. To this situation, it is important to add the early specialization, high expectations, and constant pressure in gymnastics environments that make the central pain factors in this population important [14], such as pain catastrophizing.

Therefore, this study aimed to describe the injury prevalence, pain existence, and pain catastrophizing level among gymnasts, according to their disciplines and training characteristics. We hypothesized that pain catastrophizing levels have a positive association with pain and injury prevalence, being higher in some gymnastics disciplines and with higher training volume.

## 2. Materials and Methods

### 2.1. Design

An observational study was designed. During the final part of the 2020–2021 competitive season (from May to July 2021), all Spanish gymnasts were invited to voluntarily and anonymously participate in this study by emailing their regional or national gymnastics federation. In the email, an explanation of the study was provided, and participants were asked to complete an anonymous online questionnaire to collect information about their sports characteristics, current level of pain, injuries, and pain catastrophizing level. Participants accessed the questionnaire through the JotForm platform (San Francisco, CA, USA). The STROBE guidelines were taken into account for this study [16].

### 2.2. Participants

The sample size was calculated using the GRANMO application version 7.12 (Barcelona, Spain) [17]. Accepting an 80% statistical power and a 5% significance level, 132 participants were required, with a correlation coefficient of 0.27, similar to previous studies [6,11]. As the inclusion criteria, all participants had to be at least 12 years old and have 2 years of experience in their gymnastics discipline. Prior to the start of the questionnaire, an explanation of the study was included, and informed consent was signed by the gymnasts or their legal guardians or parents in the case of minors. The Ethical Committee of the local university approved this study. Although 178 gymnasts from different gymnastic disciplines voluntarily participated in this study, 4 participants decided not to finish the questionnaire due to personal reasons. Therefore, 173 gymnasts participated in the study.

### 2.3. Online Questionnaire

The online questionnaire was designed to be anonymous and comprised four parts. The first part contained questions to describe the anthropometric, sociodemographic, and sports characteristics of the sample: age (years), sex, height (m), weight (kg), gymnastics discipline, sports experience (years), weekly training volume (hours per week), and level of competition (elite or not elite, depending on whether the gymnasts participate in international and national finals or in autonomic and national—not finals—competitions, respectively) [18]. The second part contained questions to describe sports injuries sustained during the current season: type of injury, body region, and timing. We informed the gymnasts that an injury was considered as that sustained in relation to sports or exercise with a consequent disruption in sports or exercise for at least seven days [19]. The third part of the questionnaire contained items to describe the level of pain according to the Numerical Rating Scale (NRS), an 11-point numerical tool for reporting pain, from 0 (no pain) to 10 points (worst pain imaginable) [6]. Level of pain was categorized as mild (from 1 to 3 points), moderate (from 4 to 7 points), and severe (from 8 to 10 points), as suggested by similar studies [20]. When pain existed in more than one anatomical location, it was collected considering the highest level of pain as peak pain [21]. We provided gymnasts with the following definition to identify pain: “Any pain involving muscles, tendons, and joints that occurs in a manner closely related to the specific sports practice, and that recurs in a cyclical way following the usual gymnastics sessions, in the absence of specific traumas that can justify it” [2]. The fourth part was designed to evaluate pain catastrophizing level by using the Pain Catastrophizing Scale (PCS). The PCS has 13 4-point Likert items to assess the frequency of catastrophizing thoughts about the pain experience of athletes [6]. Concretely, the PCS evaluates rumination, helplessness, and magnification, with the maximum value at 50 points. Although cut-off values are not available for the sports population [10], a score of >30 points means a high level of catastrophizing in chronic pain patients [22].

### 2.4. Statistical Analysis

Mean and standard deviation were obtained for numerical variables, and frequencies or percentages were obtained for categorical variables. The Pearson correlation was used to evaluate the relationship between PCS score, level of pain, training volume, age, and sports experience with interpretation according to these thresholds: small (r = 0.1), moderate (r = 0.3), large (r = 0.5), very large (r = 0.7), and extremely large (r = 0.9) [23]. Student’s *t*-test was used to obtain differences in PCS according to sex, gymnastics discipline, and level of competition. Analysis of variance (ANOVA) was employed to evaluate the differences between injured and noninjured gymnasts in the current season, as well as to evaluate the differences between gymnasts with current pain and without pain. Cohen’s d effect sizes (ES) were calculated to determine the magnitude of the differences between groups, interpreted as small (d = 0.2), medium (d = 0.5), or large (d ≥ 0.8). Additionally, 95% confidence intervals (95%CIs) were calculated for pain catastrophizing, injuries, and pain. The chi-square test was used to obtain differences in categorical variables among gymnastics disciplines. We used International Business Machines (IBM) SPSS statistics, version 22.0 (Chicago, IL, USA), and statistical significance was set at *p* ≤ 0.05.

## 3. Results

Since aerobic, acrobatic, and trampoline disciplines had a very low number of respondents (*n* = 5, *n* = 4, and *n* = 4, respectively), they were not considered in the data analysis (Table 1). Therefore, 160 gymnasts were included. Artistic gymnastics reported significantly higher training volume (hours per week) than the rest of the athletes (*p* = 0.006). No other significant differences were shown among gymnastics disciplines (*p* > 0.05).

**Table 1 healthcare-10-00890-t001:** Sociodemographic characteristics, injuries, pain level, and catastrophizing level for all participants.

	Total (*n* = 160)	Artistic (*n* = 59)	Rhythmic (*n* = 101)
Sex (% females)	93.8	84.7	99.0
Age (years) ^a^	16.9 (3.0)	16.8 (3.5)	16.9 (2.6)
BMI (kg/m^2^) ^a^	20.3 (2.3)	20.8 (2.4)	20.0 (2.3)
Experience (years) ^a^	9.8 (4.1)	9.6 (5.0)	9.9 (3.5)
Competition level (% elite)	16.9	28.8	9.9
Training volume (h/wk) ^a^	17.7 (12.6)	21.0 (15.3) *	15.6 (9.8)
Injured in this season (%)	50.0	50.8	49.5
Injuries (n)	106	39	67
PCS (total score) ^a^	21.4 (10.1)	20.5 (9.4)	21.9 (10.4)
PCS (rumination) (score) ^a^	8.2 (4.0)	8.0 (3.8)	8.4 (4.1)
PCS (helplessness) (score) ^a^	8.2 (4.6)	7.6 (4.2)	8.6 (4.8)
PCS (magnification) (score) ^a^	5.0 (2.8)	4.9 (2.9)	5.0 (2.8)
Pain existence (%)	74.4	78.0	77.2
Pain in 1 district (%)	30.0	25.4	32.7
Pain in 2 districts (%)	27.5	23.7	29.7
Pain in ≥3 districts (%)	16.9	20.3	14.9
Peak pain (0–10 score) ^a^	4.4 (3.0)	3.9 (2.9)	4.6 (3.0)

BMI: body mass index; h/wk: hours/week; PCS: pain catastrophizing scale; ^a^ Values are given as mean (standard deviation); * significant differences compared to the rest of gymnastics disciplines (*p* < 0.05).

Regarding injuries, 80 of the 160 gymnasts (50.0%) reported having sustained 106 injuries during the 2020–2021 season. Of the injuries, the body locations most frequently injured were the ankle (25.5% of all injuries, 16.9% of all gymnasts), knee (14.2% of all injuries, 9.4% of all gymnasts), and low back (10.4% of all injuries, 6.9% of all gymnasts) (Figure 1C) (Appendix A). According to gymnastics disciplines, 49.5% of rhythmic gymnasts and 50.8% of artistic gymnasts sustained an injury during the current season, both in the ankle, being the most frequently injured location (23.9 and 28.2% of all injuries, respectively; 15.8% of rhythmic gymnasts and 18.6% of artistic gymnasts suffered ankle injuries) (Figure 1A,B).

In terms of pain characteristics, 119 gymnasts (74.4%) reported having pain in at least one anatomical location at the time of evaluation (mild pain, 12.6%; moderate pain, 66.4%; and severe pain, 21.0%), with an average peak pain level of 4.4 ± 3.0 points. The most frequent painful locations were the low back (35.8% of all gymnasts), knee (19.1% of all gymnasts), ankle (16.2% of all gymnasts), and wrist (9.8% of all gymnasts) (Figure 1C). According to gymnastics disciplines, 77.2% of rhythmic gymnasts and 69.5% of artistic gymnasts had pain in any location (Table 1). Although the most frequent painful location was the low back among rhythmic (46.5%) and artistic gymnasts (25.4%), significant differences were found between both disciplines in the low back (*p* = 0.003) and wrist (4.0 vs. 22.0%, respectively, *p* = 0.011) (Figure 1A,B). Appendix A shows detailed data on the percentage of gymnasts with injuries and pain, according to anatomical location and disciplines.

Regarding catastrophizing level, the average value of the total score for all gymnasts was 21.4 ± 10.1 points. No significant differences were observed between rhythmic and artistic disciplines in total score or any of the values regarding rumination, helplessness, or magnification (*p* > 0.05).

When exploring training characteristics in relation to injuries, pain, and catastrophizing, we observed that the injured participants performed a statistically significant higher volume of training in terms of hours per week than uninjured participants (19.9 ± 13.4 vs. 15.4 ± 11.5; *p* = 0.026; d = 0.36; 95%CI = 0.05, 0.67). Gymnasts with current pain were significantly older (17.3 ± 3.0 vs. 15.5 ± 2.3 years old; *p* = 0.001; d = 0.77; 95%CI = −3.2, −0.8) and had more sports experience than those without pain (10.4 ± 4.1 vs. 8.0 ± 3.9 years; *p* = 0.001; d = 0.59; 95%CI = 0.41, 1.14).

Gymnasts who sustained an injury during the current season showed significantly higher PCS values than uninjured gymnasts (24.9 ± 9.4 vs. 17.8 ± 9.4; *p* < 0.001; d = 0.32; 95%CI = −10.0, −4.1). The Pearson correlation showed a higher peak pain intensity, higher number of painful locations, (r = 0.75; *p* < 0.001), and higher PCS score (r = 0.42; *p* < 0.001). No other differences or associations were found for the rest of the variables or among disciplines (*p* > 0.05).

When only female gymnasts were considered in the data analysis (*n* = 150), significant differences and correlations previously mentioned remained intact (*p* < 0.05) (Appendix A).

## 4. Discussion

The main findings of the present study showed that the prevalence of injuries and pain is extremely high among gymnasts, both associated with catastrophizing level. While the ankle was the most frequently injured region, the low back was the most frequently painful location. After exploring related factors, training volume in terms of hours per week, age, and sports experience were related factors to consider. Although these results were observed regardless of gymnastics disciplines, rhythmic gymnasts had a higher prevalence of low back pain, while artistic gymnasts suffered more frequent wrist pain.

Although we observed a high injury prevalence, our results are lower than those observed by similar studies [24,25,26]. These values may be due to differences in competitive level and injury concept. While we considered injury as an interruption of at least 7 days, other authors determined injury when it requires time loss of training or competition or medical attention, regardless of time loss [25]. In line with most studies, the most affected anatomical regions were the ankle and knee, closely followed by the wrist [14,24,25,26,27]. The high impacts derived from floor routines and landing, in addition to frequent wrist extension positioning and hand grips, would explain these results [25,28]. Since these physical demands are similar for both artistic and rhythmic gymnastics disciplines, we did not observe significant differences in the anatomical location of injuries between gymnasts from different disciplines.

Regarding the prevalence of pain, our results are in line with previous studies, which ranged from 72% [21] to 82.3% [2] of gymnasts who experienced pain. As previous studies have also demonstrated, pain existence is normalized during the training routine of gymnasts [3]. While the ankle was the most frequently injured region, the low back was the most frequently painful location. Despite the high prevalence of low back pain, mainly in rhythmic discipline, gymnasts did not recognize it as an injury because it did not require time loss [14,21]. Although extreme back movements are required in all gymnastics disciplines, rhythmic gymnasts performed more training routines of a repetitive nature [29]. For this reason, Paxinos et al. reported more overuse-type injuries in rhythmic compared to artistic gymnastics [29]. However, our results are contrary to a recent meta-analysis that suggested artistic gymnastics as a discipline more prone to low back pain compared to rhythmic gymnastics [30]. We also found significant differences in wrist pain prevalence, which was higher in artistic gymnasts compared to rhythmic gymnasts. Wrist pain has been documented as a common musculoskeletal pain in gymnastics [2], named “gymnast’s wrist”, to a specific injury in this region as a consequence of load applied during upper-extremity weight-bearing [31,32]. Previous literature has established a clear factor related to wrist pain: training volume. Our results completely agree with these studies because, in our study, the cohort of artistic athletes displayed the highest volume of training, with significantly more hours of training compared to the rest of the disciplines [2,32]. Therefore, we confirm that volume of training is an important aspect to monitor.

In terms of catastrophizing level, we observed higher values than those described by similar studies that evaluated sports populations. These studies reported catastrophizing levels ranging from 7 to 17 points, while we observed 21 points on average, regardless of gymnastics discipline [6,11,22]. This disagreement could be explained because these previous studies did not evaluate gymnastics athletes, whose physical and psychological demands are especially high [4]. Consistently, we observed high values of rumination, helplessness, and magnification, once again without differences among gymnastics disciplines. These results were directly related to peak pain, highlighting the necessity to help gymnasts cope with pain-related difficulties. The potential role of pain catastrophizing in central sensitization and the consequent emotional affection affect injury rehabilitation and sports performance [33,34]. It is important to mention that, although the cut-off to diagnose catastrophism is established at 30 points, it was defined in a nonathletic population [20]. In gymnastics, monitoring catastrophizing level and establishing reference values should be specifically considered due to the high musculoskeletal and social pressure at a young age, which may result in an extremely high prevalence of injuries and pain that are often overlooked.

We observed a significant correlation of pain existence with sports experience and age. In line with this, a review with meta-analysis observed that increased age and extended sports experience could differentiate gymnasts with low back pain from those without [30]. As Farì et al. proposed, longer sports experience supposes higher biomechanical overload and stress that, after several seasons, may result in overuse injuries and pain [2]. In fact, we observed that gymnasts who had sustained an injury during the current season had performed a higher training volume per week than those uninjured. In accordance, a previous study even suggested the importance of the threshold of training intensity as a key factor to consider in the development of frequent injuries in gymnastics such as those occurring in the wrist region [32]. In that study, gymnasts with wrist pain trained more hours per week and competed at a higher level than those without pain.

Pain existence and recent injuries were correlated with pain catastrophizing in our study. Gymnasts who had sustained an injury during the season reported a higher level of pain and pain catastrophizing. Taking into account the potential role of pain catastrophizing in central sensitization [33], high scores could affect the efficacy of the rehabilitation approach during injuries and the sports performance of athletes, especially female athletes [34]. This factor is an important aspect to monitor and manage in a sport characterized by high musculoskeletal and social pressure at a young age, with an extremely high prevalence of injuries and pain that are overlooked. Thus, prospective studies are needed to evaluate the influence of catastrophizing level on the rehabilitation process and sports performance.

The present study had limitations. As the most important limitation, we used a retrospective and self-administered online survey. Thus, data were collected without medical monitoring. This may have led to inaccurate responses. To this aspect, it is important to add the pandemic situation during which data were collected, which could have affected the results. The retrospective characteristic of our study makes it difficult to explore the role of other medical/physiological factors, or even aspects that could affect the results before the existence of pain and/or injuries. Thus, we were not able to express the injury results according to exposure to risk in training and competition. Secondly, the sample size regarding gymnastics from acrobatic, aerobic, and trampoline disciplines was small, even for these gymnastics disciplines that are not so widespread. Thus, this study did not consider these gymnastics disciplines in the data analysis. It is recommended to design specific studies with a larger sample size to provide representative data and thus reportable results according to these disciplines. Similarly, the percentage of male gymnasts was very small in our study. Although we did not observe any disturbance of results after completing additional data analysis by removing male gymnasts, future studies are recommended to specifically explore the results when considering anatomical and biomechanical differences with females.

According to the results observed in our study, new approaches are needed to introduce pain coping skills in gymnasts to reduce catastrophizing levels. These approaches should be specifically designed attending to nonmodifiable factors related to pain and injury prevalence such as age and sports experience. At the same time, modifiable factors such as volume training should be managed in order to reduce pain and injuries and thus the catastrophizing level of gymnasts. The challenge should focus on maintaining or even improving sports performance while reducing the prevalence of pain and injuries in gymnasts.

As a novel aspect, our results add pain catastrophizing level as a modifiable factor to monitor due to its association with pain and injuries among gymnasts. Although future studies should determine the predictive value of pain catastrophizing level to prevent pain or injuries, it provokes feelings such as rumination, helplessness, and magnification that make pain the experience of gymnasts difficult to manage. Identifying and monitoring pain catastrophizing values would be the first step to designing educational interventions to help gymnasts manage pain.

## 5. Conclusions

The existence of pain and injury prevalence is extremely high among rhythmic and artistic gymnasts, increasing catastrophizing experience. The ankle was the most frequently injured region, while the low back was the most frequently painful location. While a high volume of training could facilitate injuries, the accumulation of sports experience and age were related to pain experience. These results were observed in both rhythmic and artistic gymnasts, regardless of discipline, but rhythmic gymnasts had a higher prevalence of low back pain, while artistic gymnasts experienced more frequent wrist pain.

In practical applications, health and sports practitioners should be attentive to training volume and monitor catastrophizing level and pain existence to identify high values that could facilitate injury development or interfere with the rehabilitation process and performance level. In this process, age, sports experience, and specific pain or injury prevalence according to gymnastics discipline should be considered.

## Figures and Tables

**Figure 1 healthcare-10-00890-f001:**
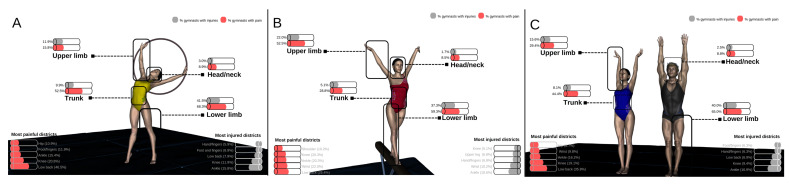
Prevalence of injuries and pain according to the anatomical district affected. (**A**) Male and female rhythmic gymnasts; (**B**) male and female artistic gymnasts; (**C**) all male and female gymnasts.

## Data Availability

The data presented in this study are available on request from the corresponding author. The data are not publicly available due to privacy.

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
