# Peer review of "Injuries, Pain, and Catastrophizing Level in Gymnasts: A Retrospective Analysis of a Cohort of Spanish Athletes"

_healthcare, 2022, doi:10.3390/healthcare10050890_

Round 1

Reviewer 1 Report

Thank you for allowing me to review this manuscript.

This study is a retrospective study which aims to describe injuries, pain and catastrophizing levels of gymnasts, according to their discipline and training characteristics.

This work is clear and well written. However, it could be improved if the results of the Pain Catastrophizing Scale (PCS) were more described in the results section and discussed as this assessment represents the greatest originality of the study.

An entire section should show the results of the CSSP in detail by district either in the form of a paragraph or a table, before the correlations are presented (line 154). A table showing these results by rumination, helplessness, and magnification for instance would be more informative than the lines in Table 1 concerning bread in 1, 2 or 3 district. The discussion could then be more powerful when it comes to the catastrophizing level.

A limitation reported by authors is the small number of certain gymnastic discipline as Trampoline, Acrobatic and aerobic. Were the results influenced by the disciplines? Perhaps the exclusion of these athletes would not be a problem given that it represents only 9 subjects, that it do not exist elite level and that pain in 1 or ≥ 3 districts is not represented. The maintenance of these 3 under-represented disciplines makes the conclusion of the work lose strength.

More in detail,

Line 67, can you add a reference for the STROBE guidelines please.

Line 97, which PCS version was used for this study: adult or child version? How to justify using only one or both versions for your population aged form 12 years (inclusion criteria) with a mean of 17 to 22 years depending from the discipline (Table 1). If the authors remove the three underrepresented disciplines, a child or adult version could be justified because the average age of trampoline athletes is 22 years while for other disciplines, the age is less than 18 years.

In table 1, the Pain Peak corresponded to the Numerical Rating Scale? Pain in 1 or other district is not informative and not described enough in the method section.

Line 210 to 219, this discussion chapter should be developed from the more detailed catastrophizing results and then from the associations found with the different factors studied.  A distinction could be made with respect to modifiable and non-modifiable factors to raise readers' awareness.

In the limitation chapter, a line (after line 222) should be added on the fact that the retrospective method could not express the injuries results according to exposure to risk in training and competition as required for other sports evaluated according with a prospective method.

If under-represented disciplines are removed, (line 222-224), the conclusion of the work will be more robust.

Author Response

Dear Reviewer,

Thank you for your helpful comments and your proposal to resubmit our paper entitled: “Injuries, pain and catastrophizing level in gymnasts: a retrospective analysis on a cohort of Spanish athletes”, pending the correction of these some major issues.

Authors are very grateful for the useful comments and time spent. The modifications have been marked up using the “Track Changes” function such that any changes can be easily located. Also, the replies to reviewer' comments are included, point by point, by explaining the details of the revisions, in bold and red format, in the attached document.

We thank you for your consideration and hope that our responses will come up to your expectations.

Yours sincerely,

The authors

Reviewer 2 Report

1 Without belittling the scientific merit of the team of authors, we believe that the conclusions they obtained are to a certain extent obvious: a gymnast who has a greater number of training sessions will have more injuries, it is unlikely otherwise! What then is the scientific idea of the study: to reduce pain and injury, let the gymnast train less?

2 The next point that reduces the scientific component is the low objectivity of the results: they talk about injuries, pains based on the information from the survey (answer choices are very subjective and situational for each person, including gymnasts), perhaps the data should be rechecked on the basis of medical (or physiological) parameters.

Our comments are only debatable and do not reduce the overall favorable impression of the article submitted for review!

Author Response

Dear Reviewer,

Thank you for your helpful comments and your proposal to resubmit our paper entitled: “Injuries, pain and catastrophizing level in gymnasts: a retrospective analysis on a cohort of Spanish athletes”, pending the correction of these some major issues.

Authors are very grateful for the useful comments and time spent. The modifications have been marked up using the “Track Changes” function such that any changes can be easily located. Also, the replies to reviewer' comments are included, point by point, by explaining the details of the revisions, in bold and red format in the attached document.

We thank you for your consideration and hope that our responses will come up to your expectations.

Yours sincerely,

The authors

Reviewer 3 Report

The authors of this article described an interesting topic, injuries, and pain in Spanish gymnasts. Despite a large amount of data and practical value, they made some inaccuracies.

Introduction

A short introduction, does not address all aspects of the study, e.g. types of injuries in gymnastics in the light of the literature.

Material and Methods

It is not entirely clear in what area the study was performed and how the questionnaire was sent.

The questionnaire study was done during a pandemic, could it have influenced the results?

Results

Is the first sentence in this section necessary, the table is signed...

The inclusion of groups with only a few respondents seems unjustified.

Trampoline (n = 4), Acrobatic (n = 4), Aerobic (n = 5).

Because of this, some of the results diverge strongly from the average, which may change it. Moreover, the description also shows the lower value of these groups.

It may be advisable to do specific studies for them.

The percentage of male gymnasts is small and, due to anatomical differences in body structure, may affect the disturbance of the results.

Discussion

Well-conducted, in some aspects there is a perceptible lack of continuity of thought.

Conclusions

Confirms assumptions from previous studies, what novelty does this research bring to the scientific knowledge?

Author Response

(The authors gave the same response as above.)

Round 2

Reviewer 3 Report

Thank you for sending corrections for the review, the article has improved thanks to this work. However, the question remains for me that the study confirms the assumptions of different studies, what does this study add to scientific knowledge?

The statement "At the same time, modifiable factors like volume training should be managed in order to reduce pain and injuries, and thus, catastrophizing level of gymnasts. The challenge should be focused on maintaining or even improving sports performance, while reducing prevalence of pain and injuries in gymnasts." is quite obvious, expected, confirmed by many studies, and is not new.

Author Response

Dear Reviewer,

Once again, thank you for your new comments and your proposal to resubmit our paper “Injuries, pain and catastrophizing level in gymnasts: a retrospective analysis on a cohort of Spanish athletes”, pending the correction of these some minor issues.

Authors are very grateful for the useful comments and time spent. The modifications have been marked up using the “Track Changes” function such that any changes can be easily located. Also, the replies to reviewer' comments are included, point by point, by explaining the details of the revisions, in bold format.

We thank you for your consideration and hope that our responses will come up to your expectations.

Yours sincerely,

The authors

Thank you for sending corrections for the review, the article has improved thanks to this work. However, the question remains for me that the study confirms the assumptions of different studies, what does this study add to scientific knowledge? The statement "At the same time, modifiable factors like volume training should be managed in order to reduce pain and injuries, and thus, catastrophizing level of gymnasts. The challenge should be focused on maintaining or even improving sports performance, while reducing prevalence of pain and injuries in gymnasts." is quite obvious, expected, confirmed by many studies, and is not new.

According to the reviewer’s recommendation, we have added information to highlight those novel aspects that our study adds to scientific knowledge: lines 282 – 288.